# Physiological mechanisms of muscle strength and power are dependent on the years post obtaining peak height velocity in elite juniors rowers: A cross-sectional study

**Paulo Francisco de Almeida-Neto**[1,2]ᵒ*, **Ayrton Bruno de Morais Ferreira**[2]ᵒ,
**Adam Baxter-Jones**[3], **Jason Azevedo de Medeiros**[1,2], **Luiz Felipe da Silva**[2], **Paulo Moreira Silva Dantas**[1,2], **Breno Guilherme de Araújo Tinôco Cabral**[1,2]*

**1** Health Sciences Center, Federal University of Rio Grande do Norte, Natal, Brazil, **2** Department of Physical Education, Federal University of Rio Grande do Norte, Natal, Brazil, **3** College of Kinesiology, University of Saskatchewan, Saskatoon, Canada

ᵒ These authors contributed equally to this work.
* paulo.neto.095@ufrn.edu.br (PFAN); brenotcabral@gmail.com (BGATC)

**Data Availability Statement:** The database for this study is publicly available at: https://figshare.com,

## Abstract

### Background

It is not yet known whether the years after peak height velocity (PHV) are associated with the physiological mechanisms of muscle strength and power in Juniors rowers.

### Objective

To identify the association between years post PHV (YPPHV) with muscle power and strength in Juniors rowers.

### Methods

We tested 235 Brazilian rowing athletes (male: 171, female: 64, Juniors category). We measured: power (indoor rowing over 100-m, 500-m, 2,000-m and 6,000-m) and muscle strength (one repetition maximum (1RM) test in squat, deadlift, bench press and bent row on the bench). Biological maturation was index by age of PHV. The sample was divided into groups considering YPPHV recent (2.5 to 3.9), median (2.51 to 4.9) and veteran (>4.9). We use a Baysian approach to data handling.

### Results

When compared to their peers in the recent and median post PHV groups, the male veteran group were superior in muscle power (Absolute: 100-m ($BF_{10}$: 2893.85), 500-m ($BF_{10}$: 553.77) and 6,000-m ($BF_{10}$: 22.31). Relative: (100-m ($BF_{10}$: 49.9)) and strength ($BF_{10} \geq 10.0$ in squat, bench press and deadlift), and in the female the veteran group were superior in test time (500-m, $BF_{10}$: 88.4).

under the Doi: 10.6084/m9.figshare.21528798.
Supplementary file 1 is publicly available at: https://
figshare.com, under the Doi: 10.6084/m9.figshare.
21528831. The full public access database can be
found on the CBR website (https://www.
remobrasil.com/bole-tins/boletins-tecnicos). The
protocol of this study is available at: https://archive.
org/details/osf-registrations-m8qr4-v1.

**Funding:** The author(s) received no specific
funding for this work.

**Competing interests:** The authors have declared
that no competing interests exist.

## Conclusion

In elite Juniors rowers the increasing YPPHV are associated with muscle power performance in both sexes and muscle strength performance in males.

## Introduction

Rowing is a sport in which youth athletes start participating in sports competitions later than many other sport—competitors are usually between 16 and 18 years old when they are entered into the official competitive context. This fact increases the possibility that athletes are in advanced stages of biological maturation (BM), indexed by biological age (BA), because the chronological age range (CA) suggests the attainment of puberty influences performance post peak height velocity (PHV) [1]. BM is the phenomenon corresponding to the enhancement of the biological systems of the human body [2].

It is worth noting that, in relation to CA, BA can be classified as early, average or late [3]. One way to assess BA is by estimating the age that PHV is attained and then aligning individuals by a BA rather than CA. PHV is a measure that estimates the period in which the highest growth rate in height occurs during adolescence, the adolescent growth spurt [4]. Therefore, adolescent athletes who have already attained PHV (post-PHV) show superior performance than their peers who have not yet experienced (pre-PHV) or are currently experiencing (circum-PHV) this BM event [1, 5]. This is justified because the physiology of the PHV stages are distinct, with the late PHV stages showing more efficient physiological mechanisms in relation to their peers in the development process (pre- and circum-PHV) [6].

Theoretically, regardless of whether the PHV occurred early, average, or late, upon attaining PHV (BA between -1 and 1) the post-PHV stage (BA >1), performance differences between adolescent athletes should be minimized or zero. However, athletes in the early BM stage reached maturity fully first compared to their average and late maturity peers [7]. It is known, that in fully mature rowers, those who reached the Tanner V stage (i.e., final stage of sexual maturity) at a younger chronological age, continue to have higher lean mass and aerobic power than their peers who reached the same stage at an older chronological age [8].

Thus, it is possible that subjects who reached the post-PHV stage at a younger chronological age (early) continue to have advantages in physical performance compared to their peers who reached the same stage at an older chronological age. Previous studies have pointed out that reaching maturity (assessed by skeletal age and PHV) are associated with and influence increases aerobic and anaerobic strength and power performance in adolescent athletes [1, 9–11]. Given this conjecture, it is justifiable to verify the relationship of years after PHV (YPPHV) with strength and muscle power performance in rowers.

Analyzing the association of YPPHV may aid exercise prescription in pediatric sport, particularly during long-term athlete training. Recently research that seeks to understand the factors involved with exercise prescription for pediatric athletes in the short and long term has been encouraged [12]. This encouragement was grounded in the 2021 Olympic games (in Tokyo, Japan), where athletes aged 12, 13, and 14 years competed in table tennis, skateboarding, and swimming respectively [12].

Given such a perspective the objective of the present study is to identify association between years after obtaining, of PHV with specific performance characteristics in rowing and the performance of muscular strength of upper and lower limbs of both sexes of Brazilian Junior

rowers. The present study hypotheses that increasing years after obtaining of PHV are related to the performance in Brazilian Juniors rowers of both sexes.

## Methods

This is a cross-sectional study with a sample of 235 rowers (Brazilians, both sex, age: 17.0 ± 1.1, category Juniors). To determine the a priori sample size we made a calculation using the software G*Power (Version 3.0; Berlin, Germany) considering a standard α of 0.05, a β of 0.80 and the effect size (0.575) of a previous study [9] that evaluated the association between power in indoor rowing and maturation in rowers of both sexes in a post-PHV maturation stage. Thus, a minimum sample size of 18 subjects per group was indicated (power sample: 0.81).

To select participants for the present study, we used data available in an open-access database of the Brazilian Rowing Confederation (CBR) regarding the National Rowing Evaluation System (SNAR). The database contains information on anthropometrics (height, wingspan, and body weight), indoor rowing-specific tests (100-m, 500-m, 2,000-m, and 6,000-m), upper limb muscle strength tests (bent row on the bench and bench press), and lower limb muscle strength tests (squat and deadlift). These tests were performed annually on nationally ranked athletes.

Initially, we had access to information from 2,870 rowing athletes of both sexes and all categories. We screened athletes from the Junior category (up to 18 years of age 573 Junior rowers. To elect the participants for the present study, we considered for the final screening that the data should contain the following complete information: (i) Performance in indoor rowing of 100 m, 500 m, 2. 000 m, and 6,000 m. (ii) Rest and peak heart rate during performance at 100-m, 500-m, 2,000-m, and 6,000-m distance. (iii) Upper limb strength and lower limb strength. (iv) Height and body weight. 235 Junior rowers fulfilled these inclusion criteria (171 males and 64 females).

### Study design

Twenty-four hours after the anthropometric evaluations, indoor rowing tests were performed in the following sequence: 100-m, 500-m, 2,000-m, and 6,000-m, with 24-hour intervals between tests. During the indoor rowing tests, exercise intensity was controlled by peak heart rate, which had to be > 80% of maximum heart rate. Forty-eight hours after the indoor rowing tests, muscle strength tests were performed in the following sequence: bench press, squat, bent row on the beach, and deadlift, with 24h intervals between tests.

We emphasize that we have contacted the CBR for information about the above procedures. Prior to all evaluations, athletes must perform a maximal stress test to determine their maximum heart rate and an echocardiogram examination to analyze the structure of the heart tissue. We emphasize that the CBR did not provide us with the results of these tests. Both tests were performed by a medical professional specializing in cardiology. Athletes can only participate in SNAR evaluations if they obtain verified medical authorization in writing. In the week after the medical clearance, the athletes were submitted to the tests mentioned above.

### Ethics and registration

The present study used a public domain database, which exempts the present study from analysis by a local ethics committee. However, for ethical reasons we excluded from the database any information that could be used to identify any of the athletes (CBR registration, name, surname and location). And it is publicly available in the Open Science Framework Registries platform (Doi: 10.17605/OSF.IO/M8QR4). To structure the present research, we followed the recommendations of the STROBE reporting guide for observational studies [13].

## Blinding

This study was blinded by the raters, the participating athletes, and the person responsible for data processing. The CBR raters were not aware of the present study when they collected the SNAR data and the athletes were not aware of the objectives of the present research. For data processing, the variable years post peak height velocity (YPPHV) was masked. Subsequently, the masked database was sent to a blinded researcher who had no prior access to the source data. Thus, the data analysis could not be manipulated to favor the initial hypothesis of the present study.

## Procedures

**Anthropometry.** The anthropometric evaluations were performed with barefoot subjects wearing only light clothing, where their body mass was measured using a digital scale; for stature, the stadiometer was used; the wingspan was measured using an anthropometric tape. All evaluations were based on the International Society of the Advancement of Kinanthropometry (ISAK) protocol [14].

**Years post peak height velocity.** Attainment of peak height velocity (PHV) was estimated from anthropometric variables. Age from PHV attainment was predicted from mathematical regression models. Details about the models can be found in the study by Moore et al. [7]. In brief, using anthropometric values (height and weight) and CA of the individuals a BA was estimated, years from PHV, where PHV = between -1 and 1 indicated pre-PHV and BA > 1 indicated post-PHV. In the present sample all values of BA were post-PHV. Subsequently, to form similar BA groups (i.e., +1, +2 and so on), we used percentiles based on tertiles as follows: below tertile 33.33 were classified as recent post-PHV (up to 2.5 years for males and up to 3.99 years for females), between tertile 33. 33 and 66.67 were classified in median post-PHV (between 2.51 and 3.99 years for males and between 4.0 and 4.99 years for females), and above tertile 66.67 were classified as veteran post-PHV (above 3.39 years for males and above 4.99 years for females).

**Heart rate.** During sprint speed and specific performance assessments, the heart rate of the participants was assessed by short-range radio wave telemetry through a Polar®-type device (model unknown, Kempele, Finland) connected to a smartphone that was in the possession of the CBR assessor.

**Sprint speed and specific performance.** Before performing the physical tests, the participants performed a specific warm-up of 15 minutes of continuous indoor rowing at a self-selected intensity. Speed was analyzed by 100 m and 500 m maximal sprint tests. Specific performance was analyzed by 2,000 m and 6,000 m tests. All speed and specific performance tests were performed on an indoor rowing machine (Concept® model-D equipped with a PM5 digital monitor, Florida, USA). The tests were performed in an air-conditioned environment (26°C). For all analyses, the indoor rowing equipment was calibrated according to Australian International Rowing Federation specifications regarding predetermined resistance factors based on sex and age group for the Juniors category (Male: 120 (Ns2/m2). Female: 110 (Ns2/m2)) [15]. At the end of testing, test time results in seconds (100-m and 500-m) and minutes (2,000-m and 6,000-m) and power output in watts (all distances) were assimilated from the equipment by a computer attached to its PM5 digital monitor.

**Upper and lower limb muscle strength.** The analysis of muscle strength of the upper limbs (bent row on the bench and bench press) and lower limbs (squat and deadlift), considered the following protocol: (i) Initially, each athlete self-selected the load (Kg) that he/she considered adequate to perform one maximum repetition (1RM). (ii) Subsequently, after performing the exercise, the athlete informed his/her subjective perception of the load (in

light, moderate and heavy). (iii) After informing his subjective perception, the CBR rater asked if he should adjust the load to a higher value; according to the athlete's response, 3% of the load was increased or reduced. Each athlete had three attempts interspersed by five minutes of passive rest to perform 1RM, in all attempts the described procedure was repeated. When the athlete reached 1RM on the first or second repetition, the test was stopped and terminated. If the athlete could not perform the movement correctly with the initial relative load, the test was stopped and the athlete was given a 10-minute passive rest period to choose a lower subjective load and try again. Details of the equipment and techniques used in the strength tests can be found in S1 File.

**Statistical analyses.** Data normality was verified by Kolmogorov-Smirnov and Z-score tests for skewness and kurtosis (-1.96 to 1.96). Subsequently, we performed a Baysian ANOVA test, assuming, a priori, running the models on a comparative basis in the null model (no difference between groups). The Bayes factor (BF) was interpreted by magnitude [16]: 1 to 3.2: Not worth more than a bare mention, >3.2 to 10: substantial, >10 to 100: strong, >100: decisive. During the analyses, we used the YPPHV-based groups as fixed effect factors and the dependent variables were the indoor rowing performance and muscle strength test results. We made inference to a single model based on the posteriori estimates. Subsequently, when the alternative hypothesis (H1) was favored substantially or strongly we considered the pointed difference worthy of post-hoc (†) [16]. Thus, we checked the point differences between the post-PHV groups by Bayesian post-hoc analysis by the null control correction. In the a priori and a posteriori specification, numerical accuracy was set manually (at 10,000 No. sampling). We used a model with fixed and uniform effects. All analyses were performed in the open source software JASP® (Version 0.16.3.0; University of Amsterdam, Holland) as recommended by Wagenmakers et al. [17], considering the error rate of 5%.

## Results

Table 1 shows the sample characterization. It can be seen that in males, the recent years from attainment of post-PHV (YPPHV) group had attained PHV two years previously, the median YPPHV group, were three years post PHV, and the veteran YPPHV group four years post PHV. In females, the recent YPPHV group had attained PHV two years previously, the median YPPHV group were t four and a half years post PHV, and the veteran YPPHV group were five and a half years post PHV. Thus, it can be observed descriptively, that subjects who were approximately 2 years post PHV (recent YPPHV) were smaller stature than their peers who were 4- or 5-years post PHV (veteran YPPHV), particularly in females. In addition, in both genders, the recent-YPPHV groups had lower body weights than their counterparts in the Median-YPPHV and Veteran-YPPHV groups. The standard error results of the performance measures are available in S2 File (Supplementary Table 1 (Table s-1)).

For males, the alternative hypothesis was favored with substantial or strong BF for absolute power produced during the 100-m, 500-m, and 6,000-m and for relative power in 100-m (See Table 2). In addition, the alternative hypothesis was favored for the 1RM tests in bench press, squat and deadlift. For females the alternative hypothesis was favored strongly only for the power produced during the 100-m test and the 500-m test time.

For males, in the power produced during the 100-m and 500-m test, post-hoc analyses showed that the veteran-YPPHV group (Absolute power in 100-m: $BF_{10}$: 3703.0. In 500-m: $BF_{10}$: 1026.8. Relative power in 100-m: $F = BF_{10}$: 36.847) and median-YPPHV group (Absolute power in 100-m: $BF_{10}$: 41.6. In 500-m: $BF_{10}$: 15.7. Relative power in 100-m: $F = BF_{10}$: 10.711) were superior to the recent-YPPHV group (See Fig 1A–1C). In the power output at 6,000-m the veteran-YPPHV group was superior to the recent-YPPHV group ($BF_{10}$: 53.6) (See Fig 1D).

**Table 1. Sample characterization.**

| Variables | | Male sex (n = 171) | | | Female sex (n = 64) | | |
|---|---|---|---|---|---|---|---|
| | YPPHV groups: | Recent | Median | Veteran | Recent | Median | Veteran |
| Participants number | | 58 | 56 | 57 | 22 | 24 | 18 |
| Age (years) | | 16.1±0.8 | 17.1±0.7 | 18.0±0.5 | 15.5±0.9 | 17.2±0.6 | 18.0±0.6 |
| PHV | | 2.0±0.5 | 3.0±0.3 | 4.0±0.4 | 3.1±0.6 | 4.6±0.3 | 5.5±0.4 |
| Weight (Kg) | | 69.7±8.4 | 71.7±8.6 | 74.4±8.4 | 58.3±4.5 | 61.7±6.9 | 67.9±7.0 |
| Stature (cm) | | 177.8±7.0 | 178.0±7.0 | 180.5±8.5 | 166.0±4.4 | 168.7±5.7 | 174.5±5.9 |
| Wingspan (cm) | | 178.4±9.5 | 181.2±15.2 | 183.1±9.1 | 172.9±9.4 | 173.8±8.0 | 175.9±5.6 |
| Resting heart rate (bpm) | | 60–9±1.1 | 58.8±0.8 | 57.0±0.8 | 60.2±1.6 | 57.1±0.8 | 55.6±0.9 |
| 100-m (sec) | | 19.1±8.6 | 20.6±14.1 | 17.0±0.8 | 32.8±3.2 | 20.5±1.0 | 19.2±0.8 |
| 500-m (sec) | | 97.2±7.1 | 94.9±5.6 | 95.1±10.0 | 113.1±6.8 | 109.1±6.6 | 104.3±5.6 |
| 2,000-m (min) | | 6.9±0.4 | 6.9±0.6 | 6.9±0.7 | 7.3±0.7 | 7.1±0.6 | 7.0±0.6 |
| 6,000-m (min) | | 21.4±2.2 | 21.1±1.8 | 20.6±2.0 | 21.6±2.7 | 21.4±2.5 | 22.5±2.2 |
| 100-m (Watts) | | 462.2±100.9 | 527.6±98.0 | 559.6±11.4 | 462.2±100.9 | 527.6±98.0 | 559.5±114.7 |
| 100-m (Watts/Kg) | | 6.6±1.3 | 7.3±0.9 | 7.5±1.2 | 5.6±0.7 | 5.4±0.8 | 5.7±0.7 |
| 500-m (Watts) | | 184.4±2.0 | 183.6±1.8 | 183.6±4.6 | 383.4±68.0 | 422.5±64.1 | 438.5±63.7 |
| 500-m (Watts/Kg) | | 5.5±1.0 | 5.9±0.8 | 5.8±0.5 | 4.9±1.1 | 5.1±1.1 | 4.8±0.7 |
| 2,000-m (Watts) | | 375.9±61.7 | 365.1±68.7 | 383.7±76.2 | 375.9±61.7 | 365.1±68.7 | 383.7±76.2 |
| 2,000 (Watts/Kg) | | 5.3±0.5 | 5.1±0.9 | 5.2±0.8 | 4.8±1.1 | 5.0±0.8 | 5.2±0.8 |
| 6,000-m (Watts) | | 272.4±72.0 | 290.2±73.0 | 325.0±84.7 | 272.4±72.0 | 290.2±73.0 | 325.0±84.7 |
| 6,000-m (Watts/Kg) | | 3.9±0.7 | 4.0±0.9 | 4.3±0.9 | 3.6±1.1 | 4.1±0.8 | 3.9±0.9 |
| Peak heart rate in 100-m (bpm) | | 176.0±5.2 | 174.6±8.0 | 176.1±9.6 | 176.3±1.4 | 175.2±11.1 | 172.9±10.2 |
| Peak heart rate in 500-m (bpm) | | 184.4±2.0 | 183.6±1.8 | 183.6±4.6 | 185.4±2.2 | 184.2±4.2 | 183.5±4.0 |
| Peak heart rate in 2,000-m (bpm) | | 193.3±2.7 | 190.5±3.4 | 189.8±4.8 | 195.3±2.6 | 189.5±3.4 | 188.3±2.2 |
| Peak heart rate in 6,000-m (bpm) | | 196.6±0.8 | 195.3±1.4 | 194.2±2.5 | 197.0±0.8 | 194.7±1.0 | 195.1±2.9 |
| 1RM in Bench press (Kg) | | 58.9±14.3 | 64.9±14.8 | 68.9±14.6 | 36.7±16.7 | 38.5±6.7 | 45.8±15.7 |
| 1RM in Rowing lyving down (Kg) | | 63.3±14.3 | 67.3±12.0 | 70.4±11.5 | 43.5±11.9 | 43.1±7.8 | 48.9±13.8 |
| 1RM in Squat (Kg) | | 85.0±19.8 | 100.6±22.3 | 104.1±23.3 | 64.3±18.8 | 66.2±13.9 | 79.8±33.6 |
| 1RM in Deadlift (Kg) | | 88.6±27.2 | 99.6±29.7 | 104.7±25.0 | 82.4±35.4 | 60.5±20.1 | 64.3±35.6 |

n: Absolut number. YPPHV: Years post peak height velocity. PHV: Peak height velocity. Sec: Second's. -m: meters. Min: Minutes. Bpm: beats per minute. Kg: kilograms. 1RM: One repetition maximum.

For the 1RM bench press test the veteran-YPPHV group was superior to the recent-YPPHV group ($BF_{10}$: 71.5) (See Fig 1E). For the 1 RM squat test the median-YPPHV ($BF_{10}$: 159.2) and veteran-YPPHV ($BF_{10}$: 2516.8) groups were superior to the recent-YPPHV group (See Fig 1F). For 1 RM in deadlift the veteran-YPPHV group was superior to the recent-YPPHV group ($BF_{10}$: 23.1) (See Fig 1G).

In the female sample, the power produced during the 100-m test in the veteran-YPPHV group was superior to the recent-YPPHV ($BF_{10}$: 10.1) and median-YPPHV ($BF_{10}$: 6.4) groups (See Fig 1H). For the 500-m performance, the veteran-YPPHV group was superior to the median-YPPHV ($BF_{10}$: 3.2) and recent-YPPHV ($BF_{10}$: 246.5) groups (See Fig 1I).

## Discussion

The purpose of the present study was to verify the association of years post PHV with the performance of Juniors category rowers. Thus, it was hypothesized that years after attainment of PHV would be associated with increased performance of Juniors rowers. Thus, the alternative

**Table 2. Comparisons between the models.**

| Variable | Models | P (M) | P(M\|data) | $BF_M$ | $BF_{10}$ | Error (%) |
|---|---|---|---|---|---|---|
| | **Male Sex** | | | | | |
| 100-m (sec) | Null (H0) | 0.50 | 0.75* | 2.91 | 1.00 | |
| | YPPHV (H1) | 0.50 | 0.25 | 0.34 | 0.34 | 0.02 |
| 100-m (Watts) | Null (H0) | 0.50 | 0.00 | 0.00 | 1.00 | |
| | YPPHV (H1) | 0.50 | 1.00* | 2893.85† | 2893.85 | 0.03 |
| 100-m (Watts / Kg) | Null (H0) | 0.50 | 0.02 | 0.20 | 1.00 | |
| | YPPHV (H1) | 0.50 | 0.98* | 49.9† | 49.9 | 0.019 |
| 500-m (sec) | Null (H0) | 0.50 | 0.82* | 4.75 | 1.00 | |
| | YPPHV (H1) | 0.50 | 0.17 | 0.21 | 0.21 | 0.02 |
| 500-m (Watts) | Null (H0) | 0.50 | 0.01 | 0.00 | 1.00 | |
| | YPPHV (H1) | 0.50 | 0.99* | 553.77† | 553.77 | 0.01 |
| 500-m (Watts / Kg) | Null (H0) | 0.50 | 0.49 | 0.99 | 1.00 | |
| | YPPHV (H1) | 0.50 | 0.51* | 1.00 | 1.00 | 0.03 |
| 2,000-m (min) | Null (H0) | 0.50 | 0.93* | 14.65 | 1.00 | |
| | YPPHV (H1) | 0.50 | 0.07 | 0.07 | 0.07 | 0.02 |
| 2,000-m (Watts) | Null (H0) | 0.50 | 0.88* | 6.99 | 1.00 | |
| | YPPHV (H1) | 0.50 | 0.12 | 0.14 | 0.14 | 0.02 |
| 2,000-m (Watts / Kg) | Null (H0) | 0.50 | 0.76* | 3.12 | 1.00 | |
| | YPPHV (H1) | 0.50 | 0.24 | 0.32 | 0.32 | 0.028 |
| 6,000-m (min) | Null (H0) | 0.50 | 0.73* | 2.77 | 1.00 | |
| | YPPHV (H1) | 0.50 | 0.26 | 0.36 | 0.36 | 0.03 |
| 6,000-m (Watts) | Null (H0) | 0.50 | 0.04 | 0.05 | 1.00 | |
| | YPPHV (H1) | 0.50 | 0.96* | 22.31† | 22.31 | 0.03 |
| 6,000-m (Watts / Kg) | Null (H0) | 0.50 | 0.27 | 0.37 | 1.00 | |
| | YPPHV (H1) | 0.50 | 0.73* | 2.69 | 2.69 | 0.04 |
| 1RM in Bench press (Kg) | Null (H0) | 0.50 | 0.05 | 0.05 | 1.00 | |
| | YPPHV (H1) | 0.50 | 0.95* | 18.8† | 18.8 | 0.03 |
| 1RM in Rowing lyving down (Kg) | Null (H0) | 0.50 | 0.27 | 0.37 | 1.00 | |
| | YPPHV (H1) | 0.50 | 0.73* | 2.65 | 2.65 | 0.04 |
| 1RM in Squat (Kg) | Null (H0) | 0.50 | 0.01 | 0.00 | 1.00 | |
| | YPPHV (H1) | 0.50 | 0.99* | 1987.55† | 1987.5 | 0.01 |
| 1RM in Deadlift (Kg) | Null (H0) | 0.50 | 0.16 | 0.19 | 1.00 | |
| | YPPHV (H1) | 0.50 | 0.83* | 5.07† | 5.07 | 0.02 |
| | **Female sex** | | | | | |
| 100-m (sec) | Null (H0) | 0.50 | 0.38 | 0.62 | 1.00 | |
| | YPPHV (H1) | 0.50 | 0.61* | 1.60 | 1.60 | 0.01 |
| 100-m (Watts) | Null (H0) | 0.50 | 0.08 | 0.09 | 1.00 | |
| | YPPHV (H1) | 0.50 | 0.91* | 11.3† | 11.3 | 0.01 |
| 100-m (Watts / Kg) | Null (H0) | 0.50 | 0.78* | 3.66 | 1.00 | |
| | YPPHV (H1) | 0.50 | 0.22 | 0.27 | 0.27 | 0.033 |
| 500-m (sec) | Null (H0) | 0.50 | 0.01 | 0.01 | 1.00 | |
| | YPPHV (H1) | 0.50 | 0.99* | 88.4† | 88.4 | 0.006 |
| 500-m (Watts) | Null (H0) | 0.50 | 0.73* | 2.72 | 1.00 | |
| | YPPHV (H1) | 0.50 | 0.26 | 0.36 | 0.36 | 0.04 |
| 500-m (Watts / Kg) | Null (H0) | 0.50 | 0.80* | 3.89 | 1.00 | |
| | YPPHV (H1) | 0.50 | 0.20 | 0.25 | 0.25 | 0.03 |
| 2,000-m (min) | Null (H0) | 0.50 | 0.73* | 2.80 | 1.00 | |

(*Continued*)

**Table 2.** (Continued)

| Variable | Models | P (M) | P(M\|data) | $BF_M$ | $BF_{10}$ | Error (%) |
|---|---|---|---|---|---|---|
| | YPPHV (H1) | 0.50 | 0.26 | 0.35 | 0.35 | 0.03 |
| 2,000-m (Watts) | Null (H0) | 0.50 | 0.26 | 0.35 | 1.00 | |
| | YPPHV (H1) | 0.50 | 0.74* | 2.81 | 2.81 | 0.01 |
| 2,000-m (Watts / Kg) | Null (H0) | 0.50 | 0.82* | 4.57 | 1.00 | |
| | YPPHV (H1) | 0.50 | 0.18 | 0.21 | 0.21 | 0.032 |
| 6,000-m (min) | Null (H0) | 0.50 | 0.77 | 3.33 | 1.00 | |
| | YPPHV (H1) | 0.50 | 0.23 | 0.30 | 0.30 | 0.03 |
| 6,000-m (Watts) | Null (H0) | 0.50 | 0.43 | 0.76 | 1.00 | |
| | YPPHV (H1) | 0.50 | 0.56* | 1.30 | 1.30 | 0.01 |
| 6,000-m (Watts / Kg) | Null (H0) | 0.50 | 0.67* | 2.06 | 1.00 | |
| | YPPHV (H1) | 0.50 | 0.33 | 0.48 | 0.48 | 0.022 |
| 1RM in Bench press (Kg) | Null (H0) | 0.50 | 0.56* | 1.29 | 1.00 | |
| | YPPHV (H1) | 0.50 | 0.43 | 0.77 | 0.77 | 0.02 |
| 1RM in Rowing lyving down (Kg) | Null (H0) | 0.50 | 0.70* | 2.31 | 1.00 | |
| | YPPHV (H1) | 0.50 | 0.30 | 0.43 | 0.43 | 0.02 |
| 1RM in Squat (Kg) | Null (H0) | 0.50 | 0.52* | 1.08 | 1.00 | |
| | YPPHV (H1) | 0.50 | 0.48 | 0.92 | 0.92 | 0.03 |
| 1RM in Deadlift (Kg) | Null (H0) | 0.50 | 0.40 | 0.70 | 1.00 | |
| | YPPHV (H1) | 0.50 | 0.60* | 1.44 | 1.44 | 0.01 |

YPPHV: Years post peak height velocity. -m: meters. Sec: Second's. Min: Minutes. 1RM: One repetition maximum. Kg: kilograms. Model Null: contains the big average of the model. Model YPPHV: adds the effect of the years post the peak height velocity event occurred. P (M): Probability of the previous model. P(M|data): Updated probabilities after the observation of the data. BF: Bayes Factor. BFM: Individual Bayes Factor for each model. $BF_{10}$: Bayes Factor for each line model relative to the null model (%): Percent. H0: Null Hypothesis. H1: Alternative Hypothesis.

*: Favored Hypothesis.

†: Post-hoc significance.

hypothesis was accepted which found that athletes veteran post PHV athletes out performance superior their peers whose attainment of PHV was more recent.

## Muscle power, biological maturation and rowing performance

In rowing, anaerobic muscle power is required during the performance of sports events, especially during the initial and final phases of the race, phases of high intensities (e.g., between 100 and 500-m) [18]. According to Moritani et al. [19] such a requirement can be justified by the recruitment of type II muscle fibers (i.e., glycolytic), i.e., the higher the intensity of exercise the greater the recruitment of this type of fiber which are responsible for the production of anaerobic muscle power. Han et al. [20] approach that type II muscle fibers have as one of their characteristics the high levels of the enzyme lactate dehydrogenase (LDH), which promotes the conversion of pyruvic acid into lactate, and the significant presence of the mAT-Pease isoform hydrolyzes more molecules of Adenosine triphosphate (ATP) than type I fibers (i.e., oxidative).

In the initial phase of the rowing race (i.e., first 500-m), due to the accumulation of hydrogen ions and the drop in intramuscular pH, there is a reduction in the activity of glycolytic enzymes (e.g., phosphofructokinase), which leads to a reduction in the anaerobic power exerted by athletes [21]. Due to this, athletes are forced to reduce the rate of strokes, thus, the production of cellular energy by anaerobic pathways is significantly reduced, and the aerobic

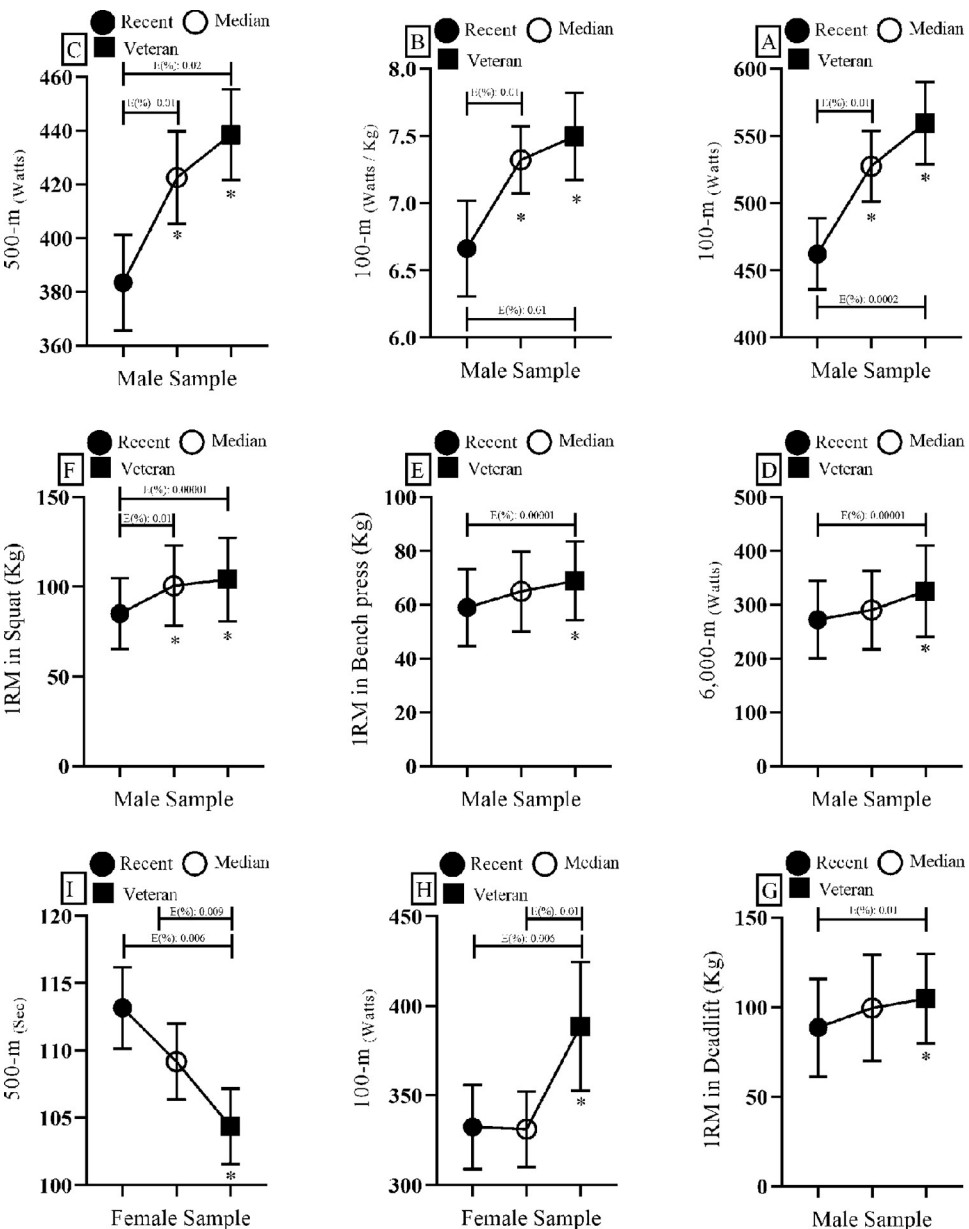

**Fig 1. Post hoc analysis of differences with substantial or strong magnitude.** E (%): Error %. -m (meters). 1RM: One repetition maximum. Kg: kilograms. * Favor of the alternative hypothesis (H1).

system becomes predominant (i.e., between 65% and 75%) [18, 22]. During this process recovery of the phosphagenic system through resynthesis of phosphocreatine will occur [23]. This contributes to the final phase of the rowing race (i.e., last 100-m to 300-m), which according to Held, Siebert & Donath [24] is where increased stroke rates occur and anaerobic metabolism becomes predominant again in cellular energy production.

In a longitudinal study that analyzed elite rowers of both sexes from the Juniors category, Almeida-Neto et al. [10] found that the advancement of biological maturation (BM) influenced the increase in power during performance in indoor rowing and in boat. Regarding muscle tissue, subjects in late BM stages point to a higher percentage of type I muscle fibers, presenting

high levels of mitochondrial density and oxidative enzyme activity [25–29]. In this sense, late-stage athletes are disadvantaged due to lower concentration of anaerobic enzymes, such as LDH, Creatine kinase (CK) and Adenylate kinase (AK) compared to their early-stage peers, who have high levels of said enzymes [6, 30].

Given this, it is expected that when reaching the final BM stage, physiological differences are balanced among adolescent athletes [1, 6]. However, previously the study conducted by Mikulic [8], followed for five years male rowers aged 12–13 years, with 10 maturing early and nine maturing late. The study analyzed sexual maturation by Tanner stages, lean body mass by anthropometry, Vo2MAX by maximal effort test performed on a treadmill, and average power by wingate test performed in indoor rowing. The primary endpoint of the study found that after five years of follow-up, the advantage of early maturers decreased in terms of lean body mass (+38% to +9%), Vo2MAX (+47% to +9%), and average power (+76% to +15%). However, although the final stage of sexual maturation was the same (post-pubertal), the subjects who reached the final stage of maturation early still had advantages over those who reached late, corroborating the present study.

Regarding the production of muscle power, Kaczor et al. [30] identified that the advancement of chronological age (CA) is determinant for the increase of anaerobic enzymes (e.g., LDH, CK and AK) and, consequently, the improvement of the efficiency of cellular energy production by anaerobic pathways. Considering the study by Benjamin [31] who found that the pace of aging is determined by biological age (BA) rather than chronological age, and the works of Malina & Bouchard [1] and Rowland [6] who detail the physiological particularities of the stages of BM in relation to CA, we can conjecture that BM is a marker of BA. Therefore, it is possible to state that subjects who reach the final stage of BM early have advanced BA compared to their peers of the same CA who reached the final stage of BM late. This thought may explain the results found by the present study regarding muscle power and years post PHV.

## Muscle strength, biological maturation and rowing performance

Rowing is a sport that interacts directly with the aquatic environment. Thus, the muscular strength exerted by the rower against water resistance is a constant reality of the sport in question [9]. Thus, muscular strength of the upper limbs, lower limbs, and trunk are directly related to the performance of rowers [32–34]. Furthermore, maturational stages have been shown to be related to strength in males, with strength gains being greater during and after PHV when compared to the pre-PHV phase [35].

This is corroborated in the present study, where individuals who were BA older (i.e. were further from attainment of PHV) (YPPHV-Veteran) performed better in the strength tests (squat, supine, and deadlift) than younger BA maturing peers (recent and median YPPHV). A possible explanation for this finding is due to the stretching-shortening cycle (SSC), which is characterized by an eccentric stretching action before a subsequent concentric shortening action [36]. SSC improves its efficiency with age. Thus, changes in the neuromuscular system during growth and maturation include increases in muscle size [11], pennation angle [37], fascicle length [38], tendon stiffness [39], motor unit recruitment[40], and preactivation [41]. These adaptations result in improved SSC performance due to elastic energy reuse, neural potentiation, and an enhanced stretch reflex contribution, mainly due to an increase in force production capabilities [42]. According to the findings of the present study, after attaining PHV the apparent residual effects of superiority in muscle strength of the BA older YPPHV-veteran group, who reached the post-PHV stage first than their peers in the YPPHV-recent and YPPHV-median groups.

In females, however, there was no difference in the performance of strength tests between the groups. It is known that adaptations in strength differ between the sexes, mainly due to

differences in circulating anabolic hormones, which are higher in boys than in girls from puberty and which directly influence the increase in muscle mass [43, 44]. The literature points out that boys gain about 7.2 kg of muscle mass per year during PHV, while girls gain only 3.5 kg per year [45]. Furthermore, PHV in females occurs on average two years earlier than in males; however, muscle mass gains decrease in girls from the age of 15, while in boys, these gains extend until the age of 20 [45]. Such a fact may explain why YPPHV-veteran female athletes show no difference in strength levels compared to their YPPHV-recent and YPPHV-median peers.

## Biological maturation, body weight and performance in rowers

Currently in rowing, competitive categories for ages >18 years are divided by body weight (lightweights & heavyweights), however, for junior categories (ages = <18 years) this body weight criterion is not considered [15], which may contribute to a physical discrepancy among adolescent athletes. Thus, it is feasible to think of strategies that may balance the differences between recent-YPPHV athletes and their median-YPPHV and veteran-YPPHV peers.

It is known that during the maturation process the gain of lean mass, power and muscle strength are greater in athletes who mature early [1, 6]. Previously it has been pointed out that rowers who reached full maturation late point out smaller size and body weight compared to their peers who reached full maturation early [8]. This suggests that the residual effect of early PHV attainment may favor body weight gain in median-YPPHV and veteran-YPPHV rowers.

Based on this assumption, the body weight can help rowers to perform a "lever" to increase the drag of the boat in the liquid environment. Considering that at the end of the "rowing act" there is synchronous extension of the ankles, knees, hips, and trunk respectively. Thus, the body weight is "thrown" back toward the bow of the boat, which can increase the speed of the boat. Thus, when prescribing training for junior rowers it is interesting to consider body weight as a determining variable for performance.

## Conjectures regarding the relationship of YPPHV to strength and power

The present study is a pioneer in looking at the relationship of YPPHV is reached to muscle strength and power in rowers. Information on the relationship or effect of reaching maturity at a younger or older chronological age is scarce. Thus, the present study conjectures that because neurophysiological maturation about strength and muscle power occurs earlier in subjects who reached biological maturity at a younger chronological age this lasts into early adulthood (18 years) compared to subjects who reached biological maturity at an older chronological age. However, this is only conjecture and more investigations are needed on the tematic.

## Limitations and suggestions for new studies

The main limitation of the present study is that we used an observational design, which made it impossible for us to prove the effect of BA on the strength and muscle power of elite rowers of both sexes in the Juniors category. We were also not able to define the individuals timing of PHV (i.e. early, average or late) We suggest that new studies should longitudinally follow rowing athletes from the pre-PHV biological maturation stage to the post-PHV stage, subsequently following post-PHV BA categories until the full adult maturity is reached. In this way we will have a more concrete answer about the existence of the effect of years from attainment of PHV. We also suggest that future research analyze the relationship of years from attainment of PHV with strength and muscle power in adolescents who do not participate in sports, so that we can verify if the gain in strength and muscle power in adolescents is related to aspects of biological maturation or training time.

## Conclusion

It is concluded that in elite Juniors rowers the increasing years post PHV attainment are associated with improved muscle power performance in both sexes and muscle strength performance in males, suggesting that the residual effects of early maturation in relation to strength and muscle power advantage still persist until the age of 18 in rowers of both sexes (between four and six years post the PHV).

As a practical application, we suggest that rowing coaches divide the Juniors category teams according to BA (years from PHV). Afterwards, plan the training of the athletes considering the years from attainment of PHV, emphasizing strength training for male athletes classified in recent-YPPHV (e.g., BA +2) and median-YPPHV (e.g., BA +3), and for power training for athletes of both sexes classified in recent-YPPHV (e.g., BA +2) and median-YPPHV (e.g., BA +3). We also suggest that nutritional interventions be performed considering YPPHV, where athletes who point recent YPPHV need a diet that enhances lean mass and body weight gain, this may help offset the body weight difference compared to their peers with median and veteran YPPHV. We hope that in this way the level of the team will be equalized.

## Supporting information

**S1 File. Details of the equipment and techniques used in the strength tests.**
(PDF)

**S2 File. Standard error results of the performance measures are available.**
(DOCX)

**S3 File.**
(PDF)

## Acknowledgments

For your support and encouragement for the development of this academic article, we thank the Federal University of Rio Grande do Norte (UFRN), the Physical Activity and Health (AFISA) research base, the Child and Adolescent Maturation Research Group (GEPMAC). The National Council for Scientific Development (CNPQ) and the Higher Education Personnel Improvement Coordination (CAPES). For the support during the production of this study we thank the researchers Matheus Dantas (ORCID: 0000-0002-1815-2251) and Radamés Medeiros (ORCID: 0000-0003-0811-2851).

## Author Contributions

**Conceptualization:** Paulo Francisco de Almeida-Neto, Ayrton Bruno de Morais Ferreira, Luiz Felipe da Silva, Paulo Moreira Silva Dantas, Breno Guilherme de Araújo Tinôco Cabral.

**Data curation:** Paulo Francisco de Almeida-Neto, Ayrton Bruno de Morais Ferreira, Jason Azevedo de Medeiros, Luiz Felipe da Silva, Breno Guilherme de Araújo Tinôco Cabral.

**Investigation:** Ayrton Bruno de Morais Ferreira, Adam Baxter-Jones, Paulo Moreira Silva Dantas, Breno Guilherme de Araújo Tinôco Cabral.

**Methodology:** Paulo Francisco de Almeida-Neto, Ayrton Bruno de Morais Ferreira, Adam Baxter-Jones, Paulo Moreira Silva Dantas, Breno Guilherme de Araújo Tinôco Cabral.

**Project administration:** Paulo Moreira Silva Dantas, Breno Guilherme de Araújo Tinôco Cabral.

**Resources:** Paulo Moreira Silva Dantas, Breno Guilherme de Araújo Tinôco Cabral.

**Software:** Paulo Moreira Silva Dantas, Breno Guilherme de Araújo Tinôco Cabral.

**Supervision:** Adam Baxter-Jones, Paulo Moreira Silva Dantas, Breno Guilherme de Araújo Tinôco Cabral.

**Validation:** Paulo Francisco de Almeida-Neto, Ayrton Bruno de Morais Ferreira, Adam Baxter-Jones, Jason Azevedo de Medeiros, Luiz Felipe da Silva, Paulo Moreira Silva Dantas, Breno Guilherme de Araújo Tinôco Cabral.

**Visualization:** Paulo Francisco de Almeida-Neto, Ayrton Bruno de Morais Ferreira, Jason Azevedo de Medeiros, Luiz Felipe da Silva, Paulo Moreira Silva Dantas, Breno Guilherme de Araújo Tinôco Cabral.

**Writing – original draft:** Paulo Francisco de Almeida-Neto, Ayrton Bruno de Morais Ferreira, Jason Azevedo de Medeiros, Paulo Moreira Silva Dantas, Breno Guilherme de Araújo Tinôco Cabral.

**Writing – review & editing:** Paulo Francisco de Almeida-Neto, Adam Baxter-Jones, Paulo Moreira Silva Dantas, Breno Guilherme de Araújo Tinôco Cabral.

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
