## [Decision Letter · Decision Letter 0]

30 Mar 2023

PONE-D-23-05680Physiological mechanisms of muscle strength and power are dependent on the years post obtaining Peak Height Velocity in elite Juniors rowers?PLOS ONE

Dear Dr. Almeida-Neto,

Thank you for submitting your manuscript to PLOS ONE. After careful consideration, we feel that it has merit but does not fully meet PLOS ONE’s publication criteria as it currently stands. Therefore, we invite you to submit a revised version of the manuscript that addresses the points raised during the review process.

We look forward to receiving your revised manuscript.

Kind regards,

Javier Abián-Vicén, Ph.D.

Academic Editor

PLOS ONE

Journal Requirements:

Additional Editor Comments:

I have completed my evaluation of your manuscript. The reviewers recommend reconsideration of your manuscript following major revision. I invite you to resubmit your manuscript after addressing the reviewers' comments.

Reviewers' comments:

Reviewer's Responses to Questions

**Comments to the Author**

1. Is the manuscript technically sound, and do the data support the conclusions?

Reviewer #1: Yes

Reviewer #2: Yes

2. Has the statistical analysis been performed appropriately and rigorously? 

Reviewer #1: Yes

Reviewer #2: Yes

3. Have the authors made all data underlying the findings in their manuscript fully available?

Reviewer #1: Yes

Reviewer #2: Yes

4. Is the manuscript presented in an intelligible fashion and written in standard English?

Reviewer #1: Yes

Reviewer #2: Yes

5. Review Comments to the Author

Reviewer #1: Thank you for the opportunity to review this manuscript. The research is interesting in principle. However, the manuscript is difficult to follow in parts. Especially the results are presented in an unclear way. Also the discussion seems to be a bit vague. This should definitely be revised. In addition, the visual abstract is not comprehensible in its current state. The methodology should also be revised: Were 100, 500, 2000 and 6000 tests completed on the same day? more informant about the maximal stress test is needed! please add reliability data for the concept 2 erg and testing procedure! Same for the strength testing procedure. How were the strength tests standardized? The results show that the body mass increases from recent to median to veteran. This must be taken into account in the power data. Eventually the differences found will disappear. In the figures, consistently indicate either watts or pace. Based on these points, I recommend a comprehensive revision of the manuscript.

Reviewer #2: As per notes attached, Some adaptations in the introduction, such as better identification of the problem, and in the methodology, such as a better description of the sample size, are necessary. It should still be considered to improve the conclusions and update the theoretical framework.

6. PLOS authors have the option to publish the peer review history of their article (what does this mean?). If published, this will include your full peer review and any attached files.

Reviewer #1: No

Reviewer #2: **Yes: **Felipe J. Aidar

---

## [Author Response · Author response to Decision Letter 0]

4 Apr 2023

Response to reviews

Manuscript title:

PONE-D-23-05680- Physiological mechanisms of muscle strength and power are dependent on the years post obtaining Peak Height Velocity in elite Juniors rowers?

Reviewer #1: 

1) Thank you for the opportunity to review this manuscript. The research is interesting in principle. However, the manuscript is difficult to follow in parts. Especially the results are presented in an unclear way. Also the discussion seems to be a bit vague. This should definitely be revised. 

Answer:

 We reviewed the results, they were reported as described by the study by Wagenmakers et al., (2018, DOI: 10.3758/s13423-017-1323-7), the authors address how to use the Baysian approach. In addition, we present in the figures only those results worthy of post-hoc, as suggested by Kass & Raftery (1995, DOI: 10.2307/2291091).

 Regarding discussion, we stress that our study is the first to address the relationship of years after PHV attainment with muscle strength and power. Previous studies address the influence of PHV and other biological maturation events (bone age, sexual maturation, expected height as an adult, etc.). Only one study that looked at rowers from pre-puberty to post-puberty addressed that after all subjects reached sexual maturation completely, rowing athletes who reached sexual maturity at a younger chronological age continued to have advantages over those who reached sexual maturity at an older chronological age (DOI: 0.1111/j.1600-0838.2010. 01200.x), we cite this study in our discussion (line 328 to 338). 

 In light of this, our discussion addressed the mechanisms involved with gains in muscular power and strength in relation to the characteristics of the stages of biological maturity, where the more mature the subject is the higher the levels of strength and muscular power. As pointed out in our introduction (line 94 to 96), when reaching full maturity, the differences in strength and muscle power should not be significant among mature athletes, but our study found that the advantages last until the age of 18! This may be justified because the body has had more time (biologically speaking) to adapt to the physiological changes resulting from the maturation process, we have inserted a brief topic in our discussion exposing this conjecture (line 381 to 389).

2) In addition, the visual abstract is not comprehensible in its current state. 

Answer:

Upon the reviewer's report we adjusted the visual abstract as described by Ibrahim et al., (2017; DOI: 10.1097/SLA.0000000000002277). 

3) The methodology should also be revised: Were 100, 500, 2000 and 6000 tests completed on the same day? 

Answer:

 We appreciate the reviewer's observation, there was a typo in the study design session, we have adjusted and identified the "washout" time between tests (line 144 to 151).

4) more informant about the maximal stress test is needed! please add reliability data for the concept 2 erg and testing procedure! 

Answer:

 Concept 2 is a valid device, we point out that the data are from the Brazilian rowing confederation as described in the manuscript (line 129 to 135 & 152 to 159), so we did not have access to the evaluations, only the final results, and we do not have test and retest data to infer reliability. However, we entered the standard error of the evaluations from the manuscript in a supplementary file (supplementary file 2).

 In addition, we point out that previously Smith & Hopkins (2012, Doi: https://doi.org/10.2165/11597230-000000000-00000), reported that well-trained rowers have a typical error in time performance of only ~0.5% between repeated 2000m attempts on this ergometer. Similarly, Almeida-Neto et al., (2020, https://doi.org/10.1371/journal.pone.0243157) found that for post-PHV maturing rowers, the rowing ergometer points to good reliability with performance in water (ICC = 0,897; IC 95% = [0,737; 0,962]). 

 Therefore, it is reliable and appropriate to use the concept II to track changes in physiological performance and factors affecting it, and considering that the sample is of elite national-level athletes and that the evaluations were performed by trained assessors trained by the Brazilian Rowing Confederation, the results of our study are reliable.

5) Same for the strength testing procedure. How were the strength tests standardized? 

Answer:

 The strength tests were standardized according to the Brazilian Rowing Confederation, we inserted a supplementary file with the details of the 1RM evaluations (Supplementary file 1).

6) The results show that the body mass increases from recent to median to veteran. This must be taken into account in the power data. Eventually the differences found will disappear. In the figures, consistently indicate either watts or pace. Based on these points, I recommend a comprehensive revision of the manuscript.

Answer:

 Thanks to the reviewer's remark, we have inserted analyses of the relative power (Watts/kg) (table 2), and only the differences for 100-m (male sex) remain (figure 1B).

Reviewer #2: 

General comments: As per notes attached, Some adaptations in the introduction, such as better identification of the problem, and in the methodology, such as a better description of the sample size, are necessary. It should still be considered to improve the conclusions and update the theoretical framework.

1) Title: Are presented satisfactorily.

Answer: ok.

2) Abstract: Are presented satisfactorily. I suggest adding some absolute and statistical results to help visualize the work.

Answer: We insert numerical results (Bayes Factor) in the results section of the summary (line 64 to 67).

3) Keywords: Please confirm that the keywords appear as descriptors in health sciences.

Answer: We adjust the keywords as requested.

4) Introduction: It should initially present a more general approach, gradually address the problem (gap), and then present the objective. The problem must be better identified. Mentioning that PHV and its relationship to what the study proposes is still unclear would not be a problem. I suggest continuing with the statements for and against the aforementioned so that we can naturally trigger the objectives.

Answer: We adjusted the introduction to expose the problem in a more appropriate way (line 98 a 108).

5) Methods: It should present the design of the study. A CONSORT or timeline should be presented in order to get a better view of the study design.

Answer: Our study complies with the STROBE chekinlist for observational studies (line 165 to 167).

6) The sample should be better explained with the number of subjects presented initially and then present the inclusion and exclusion criteria. As it was determined that 235 junior rowers (171 men and 64 women) would be needed or sufficient. please clarify.

Answer: We insert the description of the sample calculation in the methods (line 123 to 128).

7) Results: Are presented satisfactorily.

Answer: ok.

8) Discussion: Are presented satisfactorily.

Answer: ok.

9) Conclusion: Are presented satisfactorily. However, it would be necessary to present the practical applications of the results found.

Answer: We insert the practical applicability’s in the conclusion (line 408 to 414). 

10) References: Are presented satisfactorily. However, of the 45 references, only 15 are current and 30 have more than five years of publication. Please update the theoretical framework.

Answer: We tried to update the references, but could not! we hope that the reviewer understands that the theme of the study has not been widely addressed in the literature. The only study similar to the theme was published in 2010 (DOI: 0.1111/j.1600-0838.2010. 01200.x), in relation to the studies about the rowing characteristics, muscle fiber types in children and about anaerobic and aerobic enzymes we cited the original studies, when we tried to update with current studies, they did not contemplate what we need to replace the reference.

---

## [Decision Letter · Decision Letter 1]

15 May 2023

PONE-D-23-05680R1Physiological mechanisms of muscle strength and power are dependent on the years post obtaining Peak Height Velocity in elite Juniors rowers:  A cross-sectional studyPLOS ONE

Dear Dr. Almeida-Neto,

Thank you for submitting your manuscript to PLOS ONE. After careful consideration, we feel that it has merit but does not fully meet PLOS ONE’s publication criteria as it currently stands. Therefore, we invite you to submit a revised version of the manuscript that addresses the points raised during the review process.

ACADEMIC EDITOR:Thank you for the new version of your paper. The authors must respond to the comments and requirements of reviewer one for the paper to be finally accepted.

We look forward to receiving your revised manuscript.

Kind regards,

Javier Abián-Vicén, Ph.D.

Academic Editor

PLOS ONE

Journal Requirements:

Reviewers' comments:

Reviewer's Responses to Questions

**Comments to the Author**

1. If the authors have adequately addressed your comments raised in a previous round of review and you feel that this manuscript is now acceptable for publication, you may indicate that here to bypass the “Comments to the Author” section, enter your conflict of interest statement in the “Confidential to Editor” section, and submit your "Accept" recommendation.

Reviewer #1: All comments have been addressed

Reviewer #3: All comments have been addressed

2. Is the manuscript technically sound, and do the data support the conclusions?

Reviewer #1: Partly

Reviewer #3: Yes

3. Has the statistical analysis been performed appropriately and rigorously? 

Reviewer #1: Yes

Reviewer #3: Yes

4. Have the authors made all data underlying the findings in their manuscript fully available?

Reviewer #1: Yes

Reviewer #3: Yes

5. Is the manuscript presented in an intelligible fashion and written in standard English?

Reviewer #1: Yes

Reviewer #3: Yes

6. Review Comments to the Author

Reviewer #1: Thank you for this revision. All of the points I raised have been adequately addressed. However, it is now the case that when the results are considered relative to mass, only the differences for 100m performance are relevant. Even when viewed in absolute terms, there is significant overlap in the results, including in the 100m test performances. This raises the question of why trainers and coaches need to consider peak height velocity/maturation at all. The results seem to show a strong overlap, suggesting that these factors may not be as relevant as previously thought.

Overall, the interpretation of the results and the discussion should focus more on the implications of these findings for trainers and athletes. Why is it not sufficient to simply consider biological age? Combining biological age and mass, for example, could already explain a large portion of the variance in performance. These aspects should definitely be added or highlighted more prominently in the revised version.

Reviewer #3: Dear authors,

Congratulations for your work. Your revision has improved the quality of the document and it is now ready for acceptance.

7. PLOS authors have the option to publish the peer review history of their article (what does this mean?). If published, this will include your full peer review and any attached files.

Reviewer #1: No

Reviewer #3: No

---

## [Author Response · Author response to Decision Letter 1]

16 May 2023

Reviewer #1: 

 Thank you for this revision. All of the points I raised have been adequately addressed. However, it is now the case that when the results are considered relative to mass, only the differences for 100m performance are relevant. Even when viewed in absolute terms, there is significant overlap in the results, including in the 100m test performances. This raises the question of why trainers and coaches need to consider peak height velocity/maturation at all. The results seem to show a strong overlap, suggesting that these factors may not be as relevant as previously thought. 

 Overall, the interpretation of the results and the discussion should focus more on the implications of these findings for trainers and athletes. Why is it not sufficient to simply consider biological age? Combining biological age and mass, for example, could already explain a large portion of the variance in performance. These aspects should definitely be added or highlighted more prominently in the revised version.

Answer: In the results section, in the description of Table 1, we address the difference in body weight between the groups (lines 256 to 258). In addition, in the discussion session we inserted a topic on “Biological maturation, body weight and performance in rowers” (lines 384 to 402). Finally, in the concluding section, we gave a suggestion of practical applicability aimed at nutritional intervention to increase body weight in recent-YPPHV athletes (lines 436 to 440).

Reviewer #3: 

 Dear authors, congratulations for your work. Your revision has improved the quality of the document and it is now ready for acceptance.

Answer: Thanks’.

---

## [Editor Report · Decision Letter 2]

22 May 2023

Physiological mechanisms of muscle strength and power are dependent on the years post obtaining Peak Height Velocity in elite Juniors rowers:  A cross-sectional study

PONE-D-23-05680R2

Dear Dr. Almeida-Neto,

We’re pleased to inform you that your manuscript has been judged scientifically suitable for publication and will be formally accepted for publication once it meets all outstanding technical requirements.

Kind regards,

Javier Abián-Vicén, Ph.D.

Academic Editor

PLOS ONE

Additional Editor Comments (optional):

Congratulations for your work!, I consider that your paper can be published in its current form.
---

## [Editor Report · Acceptance letter]

26 May 2023

PONE-D-23-05680R2 

Physiological mechanisms of muscle strength and power are dependent on the years post obtaining Peak Height Velocity in elite Juniors rowers:  A cross-sectional study 

Dear Dr. Almeida-Neto:

I'm pleased to inform you that your manuscript has been deemed suitable for publication in PLOS ONE. Congratulations! Your manuscript is now with our production department. 

Kind regards, 

on behalf of

Dr. Javier Abián-Vicén 

Academic Editor

PLOS ONE